# Adherence to the Japanese Food Guide: The Association between Three Scoring Systems and Cardiometabolic Risks in Japanese Adolescents

**DOI:** 10.3390/nu14010043

**Published:** 2021-12-23

**Authors:** Masayuki Okuda, Aya Fujiwara, Satoshi Sasaki

**Affiliations:** 1Graduate School of Sciences and Technology for Innovation, Yamaguchi University, 1-1-1 Minami-Kogushi, Ube 755-8505, Japan; 2Department of Nutritional Epidemiology and Shokuiku, The National Institutes of Biomedical Innovation, Health and Nutrition, 1-23-1 Toyama, Shinjuku-ku, Tokyo 162-8636, Japan; fujiwaraay@nibiohn.go.jp; 3Department of Social and Preventive Epidemiology, Graduate School of Medicine and School of Public Health, The University of Tokyo, 7-3-1 Hongo, Bunkyo-ku, Tokyo 113-0033, Japan; stssasak@m.u-tokyo.ac.jp

**Keywords:** adolescents, blood pressure, cardiometabolic risks, diet quality, energy-providing nutrients, fasting plasma glucose level, Japan Food Guide Spinning Top, Shokuiku

## Abstract

The Japanese Food Guide Spinning Top (JFGST) indicates optimal intake of five food groups (grain, fish and meat, vegetables, milk, and fruits) and sugar and confectionaries. We aimed to investigate whether adherence to the JFGST in 8th grade junior high school students (*n* = 3162) was associated with cardiometabolic risks and how different scorings of the JFGST influenced the associations. Metabolic risks were assessed from anthropometrics, blood pressure measurements, and blood glucose and lipid profile measurements. Three types of scoring adherent to the JFGST were analyzed (10 points were given for each item with optimal intake; range: 0–60): the original scoring (ORG scoring); first modified scoring, which had no upper limits for vegetables and fruits (MOD1 scoring); and MOD2 scoring without upper limits for five dishes (MOD2 scoring). The MOD2 scoring was positively associated with dietary fiber, potassium, calcium, and vitamins. All types of scorings were associated with low glucose levels (*p* ≤ 0.001); the MOD2 scoring was associated with low systolic blood pressure (*p* = 0.001) and low cardiometabolic risk (*p* = 0.003). Our findings suggest that Japanese adolescents adherent to the JFGST had low cardiometabolic risks and should not fall below lower limits for intake of the abovementioned five food groups.

## 1. Introduction

Numerous dietary indices, including modifications and variants, have been proposed to assess overall diet quality representing nutrient recommendations, healthy dietary habits, and dietary variety [1]. People consume a combination of several foods containing various nutrients that are interrelated metabolically and functionally in one sitting. Dietary indices have been reported to be associated with health outcomes, such as total mortality and cardiovascular diseases [2]. Most dietary indices are based on dietary guidelines and food guides in the US, such as the Healthy Eating Index and the Diet Quality Index [3,4], or on a Mediterranean diet pattern, such as the Mediterranean Diet Score and the Mediterranean Diet Quality Index for children and adolescents [5,6]. Dietary patterns in these countries are different from those of Japan. Fat and carbohydrate in food supply accounted for 37.7–43.8% and 43.3–48.7% of the total energy, respectively, in 2017 in North America, Western Europe, and Australia. In contrast, the proportions of these nutritional components were 29.8–30.3% and 56.7–57.3%, respectively, in 2017–2019 in Japan [7]. Japanese people may consume less fat and more carbohydrates than people in countries where the aforementioned dietary indices are used. The longevity and relatively low morbidity of cardiovascular diseases in Japan are attributed to the Japanese diet. Appropriate dietary indices associated with Japanese health should be used to assess Japanese dietary patterns. Since high carbohydrate intake is considered a feature of the Japanese diet [8], a diet index comprising energy-providing nutrients could be applied to achieve health-promoting benefits.

The Japanese Food Guide Spinning Top (JFGST) was developed in 2005 as an educational tool to promote healthy behavior in Japanese people [9]. It was based on the Dietary Guidelines for Japanese and the Dietary Reference Intakes for Japanese, which take into account the Japanese culture, that is, considering carbohydrate intake. The JFGST is applicable for individuals aged 6 years or older on the assumption that they adhere to their estimated energy requirements. The JFGST is expected to be available in food and nutrition education (“Shokuiku”) campaign.

JFGST-based scoring has been used to assess diet quality in several reports. Adherence to the JFGST showed beneficial associations with the scores on mortality, metabolic risk factors, depression, and sleep [10,11,12,13,14,15,16,17,18,19,20]. However, scoring methods based on the JFGST vary among reports, that is, with or without upper limits of the recommended range of food groups [10,11,12]; with additional scoring items, such as energy [10,11], sodium intake [12], and red/white meat [11]; and using continuous points [12] or discrete points after rounding to whole numbers [16]. Different scoring methods may influence associations with health outcomes in different ways.

The original JFGST comprises five food groups with upper and lower limits using discrete points, and two items with upper limits, including sugar and confectionaries, and alcohol beverages. However, it does not include items on energy, sodium, or red and white meat [9]. Since there were few habitual consumers of alcohol beverages in Japanese adolescents (0.5–1.8%) [21], six items, including five food groups and sugar and confectionaries, are appropriate. Meanwhile, achieving dietary and nutritional balance is an essential goal of the JFGST [9]. In the Dietary Reference Intakes for Japanese, items with defined upper limits for preventing lifestyle-related diseases are sodium and saturated fatty acids [22]. However, these could not determine the upper limits of vegetable dishes, fish and meat dishes, milk, and fruits. Oba et al. used the original lower and upper limits of the JFGST [10], Kurotani et al. removed the upper limits for vegetable dishes and fruits [11], and Kuriyama et al. did not use the upper limits for five food groups [12]. While only the score without the upper limits was associated with favorable nutrient intake patterns in Japanese women [12], all three scoring systems showed inverse associations with mortality [10,11] and metabolic risk factors [15,17]. In addition, there are no reports that the scores were coincidentally calculated with or without upper limits of the food groups to assess the associations with metabolic factors except for one study on adults, which showed that both scorings had unexpected associations with serum cholesterols and glycosylated hemoglobin [17].

Previous reports based on the JFGST targeted adult population [10,11,12,13,14,15,16,17,18,20], except for one study that analyzed a population aged < 6 years [19], which is not a target age population of the JFGST. However, there are no reports for adolescents. Therefore, the association between metabolic risk factors and the scoring system in adolescents is still unknown. The aim of the study was to investigate the association between diet quality score of adherence to the JFGST, representing a feature of Japanese diet, and metabolic risk factors in Japanese adolescents. We examined three scoring systems to clarify the different factors that influence the associations between the scores and cardiometabolic risk factors.

## 2. Materials and Methods

### 2.1. Participants

This study was part of the Shunan Child Cohort Study described in detail elsewhere [23,24,25]. The participants were 8th-grade junior high school students from 17 junior high schools in Shunan City, Japan. From 6805 students attending any of the schools between 2006 and 2010, 6226 students participated in this study with their guardians’ consent. We excluded those with missing variables; with physician-diagnosed diseases; who had taken breakfast before blood extraction; with high plasma glucose, triglyceride, or low-density lipoprotein cholesterol levels due to possible postprandial values, or familial hypercholesterolemia; and with implausible energy intake (Figure 1). Overall, we analyzed the data of 3162 participants.

### 2.2. Dietary Assessment

Foods and nutrients were assessed using a brief-type self-administered diet history questionnaire for youths (BDHQ15y). The BDHQ15y, which assesses the consumption frequency of 63 selected food items and 17 dietary behaviors in the previous month, is a modification of the BDHQ for adults. The correlation coefficients between the estimates from the single BDHQ for adults and 16-day dietary records were 0.17–0.66 (Spearman correlation) for cereals, sugar and confectionaries, vegetables, potatoes, fruits, fish, meat, egg, dairy products, and non-alcoholic beverages [26]; 0.35–0.64 (Pearson correlation) for protein, fat, and carbohydrate; 0.44–0.66 (Pearson correlation) for sodium, potassium, calcium, magnesium, and iron; and 0.42–0.63 (Pearson correlation) for beta-carotene equivalent, and vitamin C [27]. Spearman correlation coefficients with corresponding biomarkers in adolescents were 0.26–0.31 for serum carotenoids; 0.22–0.48 for red blood corpuscle marine omega-3 polyunsaturated fatty acids [28]; 0.11–0.30 for urinary nitrogen [29]; and 0.05 for sodium; 0.11 for potassium; and 0.10 for the sodium-to-potassium ratio [30]. Plausible responders were considered to have energy intake ≥0.5 and ≤1.5 times of age- and sex-specific estimated energy requirements for low and high physical activity levels (PALs), respectively [31]. Intake of nutrients and food was adjusted using an energy density method [32]. As we could not determine the PAL of the participants, individual intake was standardized assuming energy intake equal to age- and sex-specific estimated energy requirements for moderate PAL.

Diet quality was assessed from five food dish groups (grain dishes, vegetable dishes, fish and meat dishes, milk, and fruits) and sugar and confectionaries based on the JFGST (Appendix A) [9,33]. The original score was calculated from the standardized energy-adjusted nutrient and food intake based on the serving standards of the JFGST scoring system. For five dishes, one serving (SV) corresponded to 40 g of carbohydrate for grains, 70 g of vegetables for vegetable dishes, 6 g of protein for fish and meat, 100 mg of calcium for milk, and 100 g of fruits for fruits. The number of servings was rounded to whole numbers as follows: if the value obtained was between 0.67 and <1.5, it was counted as one serving; any value between 1.5 and <2.5 was rounded off to two servings; and any value between 2.5 and <3.5 was rounded off to three servings. The rounding-off manner applies to the succeeding values (i.e., 3.5 to <4.5 and so on). The maximum point per dish was 10. When the intake was lower or higher than the optimum, a point was calculated as SV/lower limit of the optimum × 10 or 10 − (SV − the upper limit)/upper limit × 10, respectively (Appendix A). When the calculated point was less than zero, zero point was given. Optimum intake of sugar and confectionaries, which are not counted in SV, was <200 kcal giving 10 points, and points for ≥200 kcal were calculated (corresponding energy − 200)/200 × 10. When the calculated point was less than zero, zero point was given. The first modified score (MOD1 score) was the same as the original score (ORG score), except that no upper limits were set for vegetable dishes and fruits according to Kurotani et al. [10,11]. The second modified score (MOD2 score) had no upper limits for the five dishes according to Kuriyama et al. [12]. The scores of three scoring methods ranged from 0 to 60, with higher values indicating higher adherence to the JFGST.

### 2.3. Cardiometabolic Risk Factors

School nurses measured the participants’ body height and weight near 0.1 cm and 0.1 kg, respectively. Body mass index (BMI; kg/m^2^) was calculated as the weight divided by the square of height. The standard deviation score of BMI (zBMI) was calculated based on the Japanese reference in 2000 [34] using the Lambda–Mu–Sigma method [35]. Blood pressure (BP) was measured twice using auto-sphygmomanometers (HEM-707, HEM-757, or HEM-780, OMRON, Kyoto, Japan) after 5 min of sitting. Mean systolic and diastolic blood pressure (SBP and DBP, respectively) measurements were used. The participants were asked to fast 10–12 h before the blood extraction. Triglyceride (TG), low-density and high-density lipoprotein serum cholesterol (LDL-C and HDL-C, respectively), and plasma glucose levels were measured from blood specimens. Cardiometabolic risks were defined based on the definition of the International Diabetes Federation (32): zBMI ≥ 1; TG ≥ 150 mg/dL; LDL-C ≥ 120 mg/dL; HDL-C < 40 mg/dL; fasting plasma glucose ≥ 100 mg/dL; and SBP ≥ 130 mmHg and/or DBP ≥ 85 mmHg. Cardiometabolic risks were summed up as a metabolic syndrome score (MS score) ranging from 0 to 6.

### 2.4. Confounding Factors

The lifestyle questionnaire assessed movement behavior and household information. Sports activity (>2 times/week or not), TV watching (>2 h/d or not), sleep duration (h), number of siblings (1, 2, or ≥3), and single parent (yes or no) were used as possible confounders.

### 2.5. Statistical Analysis

Unless specified, variables are expressed as mean ± standard deviation or count (%). After checking for normality using quantile–quantile plots, the JFGST score points and triglyceride levels are expressed as median (minimum, maximum). Scores were categorized based on quintile levels (the lowest Q1 to the highest Q5). The median of energy-adjusted food and nutrient intake (% of total energy intake for energy-providing nutrients and amount per 1000 kcal for remaining nutrients and food groups) in each category and the ratio of intake in Q5:Q1 (%) were calculated to examine the contribution of foods and nutrients to the score. Non-parametric trend tests for food and nutrients across Q1 to Q5 were performed using the Jonckheere–Terpstra test for two-sided monotonic trends. Gaussian linear regression models were used to examine the association between the score and individual cardiometabolic risks. When the triglyceride level was used as a dependent variable, it was natural log transformed. Poisson regression models were used for the MS score because it had a skewed distribution of zero (Table 1); the effect on the dependent variable was an exponential function of a coefficient. Coefficients of both regression models were calculated as an effect of 10 points of score. R version 4.0.3 (R Foundation for Statistical Computing) was used [36], and the significance level was set at *p* = 0.05.

## 3. Results

The mean age of the participants was 13.6 ± 0.3 years, and the average BMI was 19.2 ± 2.6 kg/m^2^ (Table 1). Of 3162 participants, 27.3% had one or more cardiometabolic risk factors.

For many students, the servings of fish and meat dishes and milk were over the upper limits (70.2% and 52.3%, respectively), and some of them had zero points for these dishes (11.7% and 41.1% of total participants, respectively; Appendix A). Therefore, the median points for fish and meat dishes and milk using the scoring with the upper limits (6.7 and 5 points, respectively) increased when using the scoring without the upper limits (10 and 10 points, respectively; Table 2). For less than 10% of the students, the points for grain dishes (3.4%), vegetable dishes (9.7%), and fruits (4.3%) were above the upper limits. The points for these dishes were similar regardless of whether the upper limits were applied or not. The median JFGST scores were 32 (minimum: 6.7, maximum: 58.3) using the ORG scoring, 33 (3.7, 58.8) using the MOD1 scoring with no upper limits for vegetable dishes and fruits, and 40.9 (15, 60) using the MOD2 scoring with no upper limits for all dishes except for sugar and confectionaries.

Using the MOD2 scoring, the higher the total score, the higher the median intake of fish and meat, fish, and milk (Q5:Q1 ratios, 122–282%), whereas other scorings showed opposite trends of the medians across the quintile categories in these food groups (Q5:Q1 ratios, 62–83%; Table 3). In addition, the MOD2 scoring had higher Q5:Q1 ratios for vegetables and fruits than other scorings. In terms of nutrients, protein, sodium, and calcium intake showed opposite trends of the medians across the quintile categories in the MOD2 scoring and other scorings (Table 4); the higher score of the MOD2 scoring, the higher median intake of protein, sodium, and calcium. Beneficial health nutrients, such as dietary fiber, potassium, β-carotene equivalents, and vitamin C, had larger differences between the lowest and the highest score categories (Q1 and Q5) in the MOD2 scoring (Q5:Q1 ratios, 141–229%). Saturated fatty acids showed similar trends across the three types of scorings; Q5:Q1 ratios in the ORG and MOD1 scores were 72%, and that in the MOD2 score was 87%.

The scores of the three scorings were significantly associated with low glucose levels (all *p* < 0.001; Table 5). The Akaike’s information criterion was the lowest in the model with the MOD2 score as a dependent variable. Only the MOD2 scoring was significantly associated with low SBP (−0.81 mmHg per 10 points of the score; *p* = 0.001). The MS score was also significantly associated with the MOD2 scoring (*p* = 0.003); an increase in the score by 10 points was related to 0.89 (e^−0.12^) times in the number of risks.

## 4. Discussion

We examined three scoring types of dietary quality based on the JFGST. The contribution of each food group to the total score and the trends of nutrients and foods across the quintile categories were similar between the original and the first modified (ORG, and MOD1) scoring with no upper limits for vegetable dishes and fruits. In contrast, the contribution of fish and meat dishes and milk to the score was high in the MOD2 scoring with no upper limits for all dishes except for sugar and confectionaries. In addition, the MOD2 scoring and other scorings showed opposite trends for protein, sodium, and calcium intake, and the MOD2 scoring could differentiate the variance in dietary fiber, potassium, and vitamins compared with other scorings. All the three scoring types were significantly associated with the fasting glucose level. Notably, the MOD2 score was significantly negatively associated with SBP and the MS score.

Based on the data from the National Health and Nutrition Survey Japan, the modified score without upper limits for five dishes was negatively associated with SBP in adults ≥20 years old of both sexes and with waist circumference in women [17]. In this study, only the modified score without upper limits for five dishes (MOD2 score) was associated with SBP and metabolic syndrome. However, in a study of female dietetic students aged 18–22 years, the score was negatively associated with waist circumference and low-density lipoprotein cholesterol but was not associated with SBP [15]. Thus, the associations noted in female dietetic students were different from those in adolescents. A systematic review of diet quality indices developed for other countries showed associations with a reduced risk of cardiovascular disease and its risk factors [2]. Diet quality indices based on the JFGST were related to low all-cause and cerebrovascular disease mortality [10,11]. In adolescents, diet quality adherence to the JFGST may imply an association with low cardiometabolic risks. In particular, the MOD2 scoring without the upper limits for five dishes is a suitable index for high intake of dietary fiber, minerals, vitamins, fish, vegetables, fruits, and milk.

The higher the participants’ MOD2 score points without upper limits for the five dishes, the higher their sodium intake, whereas using other scoring methods, the lower the sodium intake. Similar to other food frequency questionnaires, the BDHQ is vulnerable to measurement errors and is weak in estimating nutrients from foods that have not been investigated. Although sodium intake estimated from the BDHQ was barely associated with urinary sodium excretion as an intake biomarker, the sodium-to-potassium ratio from the BDHQ was significantly associated with the urinary ratio [30]. Furthermore, the sodium-to-potassium ratio was more closely associated with blood pressure than sodium intake in previous studies in youths [37,38]. In this study, the range of the sodium-to-potassium ratio across quintile categories was the widest in the scoring without upper limits for the five dishes. Thus, the association between the scoring without upper limits for five dishes and SBP, as well as the MS score, may be attributed to the sodium-to-potassium ratio.

Among the tentative dietary goals of preventing life-style related diseases, tentative dietary goals for sodium, saturated fatty acids, and a balance of energy-providing nutrients have upper limits of intake. Trends of saturated fatty acids across the quintile categories were similar among the three types of scorings, but the ORG and MOD1 score showed higher saturated fatty acids in the lowest categories than the MOD2 score did; this may reflect that the participants with intake above the optimum had low score points due to upper limits. However, the ORG and MOD1 scores did not show significant associations with cardiometabolic risk factors except for plasma glucose. The intake of two energy-providing nutrients (carbohydrates and protein) corresponds to servings of grain and fish and meat dishes in the JFGST. Two of the three energy-providing nutrients (fat being the third) can determine the nutritional balance. The tentative dietary goals for the energy-providing nutrients for the prevention of lifestyle-related diseases are 13–20% for protein, 20–30% for fat, and 50–65% for carbohydrate according to the Dietary Reference Intakes for Japanese, 2020 [22]. Fat intake in the lowest categories of the three scores was above the tentative goals, but fat intake across the quintile categories was similar among the three scorings. The apparent difference in the trends across the quintile categories among the three scores was protein, although total protein is not considered a pivotal determinant of the nutritional balance for the tentative dietary goals to prevent lifestyle-related diseases [22]. Thus, nutrients other than those providing energy may determine the association between the score and cardiometabolic risks. A rice grain dish is the main staple of the Japanese diet [8,39], but this may not be a main source of health benefits.

The ORG scores with upper limits for vegetable dishes and fruits, and the MOD1 scores without upper limits had similar points for these food groups and dietary nutrient patterns. This means that the proportion of participants with vegetable and fruit intake above the upper limits was low; thus, this population could not ascertain the beneficial effects of vegetable and fruit intake in the JFGST scoring. The Health Japan 21 (the second term) program recommends a mean daily intake of vegetables ≥350 g, which represents a notable increase from 282 g in 2010. In addition, it aims to reduce the proportion of individuals who consume fruits <100 g from 61.4% in 2010 to 30% [22]. The MOD2 scoring, which showed large differences of vegetables and fruits between the lowest and highest scores, was negatively associated with metabolic risks. The MOD2 scoring results suggest limited evidence to support an intervention to increase adolescents’ consumption of vegetables and fruits from their current intake. However, rather than focusing on a single food group, such as vegetables and fruits, consuming a healthy balance of food should be considered for Japanese adolescents.

Similar to previous dietary indices associated with cardiovascular diseases rather than cancers [1,2], the JFGST was associated with cardiometabolic risk factors. The JFGST was developed in consideration of the concept of Japanese culture. Japanese terms for dishes are derived from ichiju-sansai, a common formula of Japanese cuisine, Washoku. Grain dishes, shushoku, are the main staples; fish and meat dishes, shusai, are the main dishes; and vegetable dishes, fukusai, are the side dishes [40]. The concept of Japanese dishes is disseminated through homemaking classes and nutrition education at school. The JFGST is a food-based index unlike most of other existing dietary indices [41]. The Healthy Eating Index, the Diet Quality Index, and the Mediterranean Diet Quality Score use foods and nutrients together as scoring items [3,4,5]. The Mediterranean Diet Quality Index for children and adolescents uses food intake and dietary habits, but this scoring may be influenced by culture [6]. When using the JFGST as a health promotion and education tool, adolescents can easily understand their suitable dish servings. For effective nutrition education, the dietary index should be suited to the background of the target population.

In contrast to the scoring items, the JFGST scoring uses a combination of nutrient and food intake; the servings of grain dishes, fish and meat dishes, and milk were calculated from nutrient amount, but the servings of vegetable dishes and fruits were from the food amount. It is unknown whether servings that adolescents have in mind coincide with those of the JFGST scoring. Dietary indices have three objectives: to measure absolute diet quality, to evaluate adherence to dietary guidelines, and to guide health promotion [41]. According to the third objective, Japanese youths should be educated on favorable dietary habits to ensure longevity and low mortality. Practical use of existing dietary indices for health promotion should be addressed [1,2], and the JFGST should be adopted for nutrition education interventions.

This study had some limitations. The cross-sectional design of this study could not explain temporal causality between the scores and cardiometabolic risk factors, but the participants might not have known their own cardiovascular parameter levels before answering the BDHQ15y. The design could also not explore the effect on risks appearing in later life, such as cardiovascular mortality, cancer incidence, frailty, and dementia related to healthy life expectancy. Social norms regarding a healthy diet may contribute to a reporting bias, but this may attenuate observable associations. The BDHQ15y has weaknesses, similar to other frequent food questionnaires. Intrinsic measurement errors and biases of the BDHQ15y may attenuate or mask the true associations between the score and cardiovascular risk. Nevertheless, the associations found in this study could help adolescents whose diets are assessed using the BDHQ15y to review and modify their own dietary habits with the JFGST score. Another limitation is that the study location was limited to a small part of Japan, meaning that the data collection could not capture all Japanese dietary patterns.

## 5. Conclusions

We showed for the first time that Japanese adolescents adherent to the JFGST had low cardiovascular risk. However, only the modified JFGST scoring system with no upper limits for five food groups (grain dishes, fish and meat dishes, vegetable dishes, milk, and fruits) was beneficial in alleviating cardiovascular risks; this indicates that adolescents should not fall below the lower limits for intake of the abovementioned five food groups. The JFGST can be used for adolescent health education.

## Figures and Tables

**Figure 1 nutrients-14-00043-f001:**
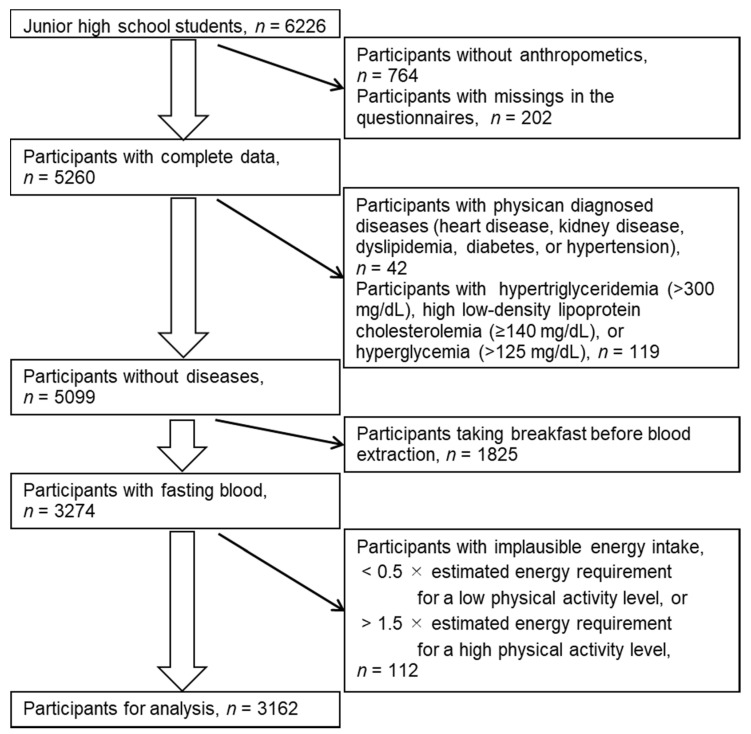
Selection of participants for analysis.

**Table 1 nutrients-14-00043-t001:** Characteristics of the analyzed participants (*n* = 3162).

Characteristics	Value
Age, years	13.6 ± 0.3
Body mass index (BMI), kg/m^2^	19.2 ± 2.6
z-score of BMI	−0.2 ± 0.9
Triglyceride, mg/dL	52 (15, 249)
LDL-C, mg/dL	88.1 ± 19
HDL-C, mg/dL	67.7 ± 13.9
Fasting plasma glucose, mg/dL	90 ± 5.8
Systolic blood pressure, mmHg	114.2 ± 11.6
Diastolic blood pressure, mmHg	68.1 ± 8.7
Energy intake, kcal/day	2220 ± 632
Sex	
Male	1627 (51.5)
Female	1535 (48.5)
MS score	
0	2298 (72.7)
1	684 (21.6)
2	150 (4.7)
3, or 4	30 (1.0)

Data are expressed as mean ± standard deviation, median (minimum, maximum), or count (%). LDL-C, low-density lipoprotein cholesterol; HDL-C, high-density lipoprotein cholesterol. The MS score (metabolic syndrome score) is the clustering of cardiometabolic risks.

**Table 2 nutrients-14-00043-t002:** Diet quality scoring based on the Japanese Food Guide Spinning Top for adolescents (*n* = 3162).

	Applying Both Upper and Lower Limits			Applying Only the Lower Limits			Applying Only the Upper Limits		
	Median	Min.	Max.	Median	Min.	Max.	Median	Min.	Max.
Grains	8.3	2	10	8.3	2	10	—	—	—
Vegetables	6.7	0	10	6.7	0	10	—	—	—
Fish and meat	6.7	0	10	10	0	10	—	—	—
Milk	5	0	10	10	0	10	—	—	—
Fruits	5	0	10	5	0	10	—	—	—
Sugar and confectionaries	—	—	—	—	—	—	0	0	10

Min., minimum; Max., maximum.

**Table 3 nutrients-14-00043-t003:** Median intake of food groups among quintile categories of the diet quality scores.

	ORG Score				MOD1 Score				MOD2 Score			
	Q1	Q3	Q5	Q5:Q1	Q1	Q3	Q5	Q5:Q1	Q1	Q3	Q5	Q5:Q1
Grain, g/1000 kcal	167	215	253	152%	170	213	251	148%	193	209	231	120%
Vegetable, g/1000 kcal	96	113	136	141%	93	113	142	153%	78	109	155	198%
Fish and meat, g/1000 kcal	125	107	104	83%	125	107	104	84%	102	107	125	122%
Fish, g/1000 kcal	31	26	24	80%	30	26	25	83%	23	27	30	132%
Meat, g/1000 kcal	39	33	31	80%	39	33	31	80%	36	32	36	101%
Milk, g/1000 kcal	132	92	82	62%	133	91	82	62%	44	102	124	282%
Fruits, g/1000 kcal	14	21	34	248%	13	22	37	281%	12	26	39	340%
Sugar and confectionaries, g/1000 kcal	119	92	62	52%	119	93	60	50%	133	97	46	35%

ORG score, original score. MOD1 (first modified) score was calculated as scores with no upper limits for vegetable dishes and fruits. MOD2 (second modified) score was calculated as scores with no upper limits for all dishes except for sugar and confectionaries. Jonckheere–Terpstra tests were used to test the trend across the quintile categories of the scores, and all trends were significant (*p* < 0.001), except for the trend of meat intake across the MOD2 score (*p* = 0.874).

**Table 4 nutrients-14-00043-t004:** Median intake of nutrients among quintile categories of the diet quality scores.

	ORG Score				MOD1 Score				MOD2 Score			
	Q1	Q3	Q5	Q5:Q1	Q1	Q3	Q5	Q5:Q1	Q1	Q3	Q5	Q5:Q1
Protein, % of total energy intake	14.8	13.8	13.5	91%	14.7	13.8	13.6	92%	13.0	14.3	15.3	118%
Fat, % of total energy intake	34.0	30.2	27.1	80%	34.2	30.1	27.1	79%	32.0	29.6	28.6	89%
SFA, % of total energy intake	11.1	9.5	8.1	72%	11.2	9.4	8.1	72%	10.0	9.3	8.7	87%
Carbohydrate, % of total energy intake	49.2	54.1	57.8	117%	49.1	54.2	57.9	118%	53.4	54.6	54.6	102%
Dietary fiber, g/1000 kcal	4.7	5.3	5.9	125%	4.6	5.2	6.1	131%	4.4	5.8	6.3	142%
Sodium, mg/1000 kcal	1918	1807	1755	92%	1902	1799	1774	93%	1761	1843	1919	109%
Potassium, mg/1000 kcal	1172	1161	1203	103%	1150	1163	1231	107%	985	1255	1386	141%
Na/K ratio	2.8	2.7	2.5	92%	2.8	2.7	2.5	89%	3.1	2.5	2.4	78%
Calcium, mg/1000 kcal	380	326	292	77%	375	325	295	79%	260	345	374	144%
Magnesium, mg/1000 kcal	119	117	123	104%	117	117	124	106%	105	127	137	131%
Iron, mg/1000 kcal	3.8	3.7	3.8	101%	3.7	3.6	3.9	103%	3.4	3.9	4.2	121%
β carotene equivalents, μg/1000 kcal	1067	1291	1578	148%	1025	1275	1687	165%	829	1528	1896	229%
Vitamin C, mg/1000 kcal	44	51	61	140%	42	51	64	152%	38	59	68	176%
Energy, kcal	2044	2130	2184	107%	2057	2127	2171	106%	2062	2149	2016	98%

ORG score, original score. MOD1 (first modified) score was calculated as scores with no upper limits for vegetable dishes and fruits. MOD2 (second modified) score was calculated as scores with no upper limits for all dishes except for sugar and confectionaries. Jonckheere–Terpstra tests were used to test the trend across the quintile categories of the scores, and all trends were significant (*p* < 0.001), except for the trend of iron intake across the ORG score (*p* = 0.413) and energy intake across the MOD2 score (*p* = 0.130).

**Table 5 nutrients-14-00043-t005:** Regression analysis of the effect of 10-point scores on cardiometabolic risks.

	ORG Score				MOD1 Score				MOD2 Score			
	β	SE	*p*	AIC	β	SE	*p*	AIC	β	SE	*p*	AIC
zBMI ^1^	−0.03	0.02	0.141	8456.6	−0.03	0.02	0.133	8457	−0.01	0.02	0.641	8459
ln (TG, mg/dL) ^1^	0.01	0.01	0.207	3601	0.01	0.01	0.249	3601	−0.01	0.01	0.285	3601
LDL-C, mg/DL ^1^	−0.59	0.44	0.172	27,559	−0.58	0.43	0.173	27,559	−0.45	0.44	0.305	27,559
HDL-C, mg/dL ^1^	−0.49	0.30	0.106	25,285	−0.57	0.30	0.057	25,284	−0.16	0.31	0.603	25,287
Glucose, mg/dL ^1^	−0.46	0.13	<0.001	19,949	−0.45	0.13	<0.001	19,949	−0.50	0.13	<0.001	19,947
SBP, mmHg ^1^	−0.09	0.25	0.710	24,120	−0.12	0.25	0.633	24,120	−0.81	0.25	0.001	24,110
DBP, mmHg ^1^	−0.02	0.20	0.936	22,542	−0.01	0.19	0.963	22,542	−0.24	0.20	0.235	22,541
ln (MS score) ^2^	−0.06	0.04	0.110	4264	−0.07	0.04	0.085	4264	−0.12	0.04	0.003	4258

^1^ Analyzed using the linear model. ^2^ Analyzed using the Poisson model, with the number of cardiometabolic risks as a dependent variable. Both regression models were adjusted for age, sex, zBMI, sports activity, TV watching, sleep duration, number of siblings, and single parent. Models for zBMI as a dependent variable were adjusted for age, sex, sports activity, TV watching, sleep duration, number of siblings, and single parent. SE, standard error of a coefficient estimate (β); AIC, Akaike’s information criteria; zBMI, z-score of body mass index; TG, triglyceride; LDL-C and HDL-C, low- and high-density lipoprotein cholesterol, respectively; SBP and DBP; systolic and diastolic blood pressure, respectively. The MS score is the clustering of cardiometabolic risks.

## Data Availability

The raw data supporting the conclusions of this article will be made available by the corresponding author without undue reservation.

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
