# Peer review of "Adherence to the Japanese Food Guide: The Association between Three Scoring Systems and Cardiometabolic Risks in Japanese Adolescents"

_nutrients, 2021, doi:10.3390/nu14010043_

Round 1

Reviewer 1 Report

The manuscript entitled „Adherence to the Japanese Food Guide: The association between three scoring systems and cardiometabolic risks in Japanese adolescents” presents interesting issue but some problems must be corrected.

Major:

Authors should be aware that they do not prepare the basic manual for students, or column of the newspaper, but a scientific paper, so they should (1) use a proper scientific vocabulary and language, (2) avoid excessive simplification. Sentences such as “Adolescents adhere to the JFGST were healthy, and it is useful for adolescents, too.” Are not acceptable.

Abstract:

Instead of what was done, Authors should present the aim of the study (e.g. “The aim of the study was…”)

Scoring systems should be briefly described.

Introduction:

Instead of what was done, Authors should present the aim of the study (e.g. “The aim of the study was…”)

Materials and Methods:

The representativeness of the studied group should be reflected.

It seems that Authors did not verify the normality of distribution and they treated all the variables as normally distributed, but based on the presented mean and SD values, it may be supposed that data for some variables were characterised by the distribution different than normal.

Authors should (1) verify the normality of distribution, (2) for normally distributed data present mean and SD values, but for the other distributions – present median, min and max values, (3) apply adequate statistical tests, that are based on the distribution.

Resuls:

For normally distributed data Authors should present mean and SD values, but for the other distributions – present median, min and max values.

Authors should apply adequate statistical tests, that are based on the distribution.

Discussion:

Authors should formulate implications of the results of their study and studies by other authors, but they should be relevant not only for Japanese readers, but rather for international auditory

Authors should formulate the future areas which should be studied (not only in Japan, but in various countries)

Conclusions:

This section is not a summary – Authors should not reproduce the previous sections in this section. Authors should briefly (2-3 sentences) formulate the most important general conclusions based on the conducted study.

Author Contributions:

Authors should be consistent with Authorship (is it FA, or AF?)

Author Response

Thank you for your review. The reviewer’s comments are valuable and have helped us to improve our manuscript.

Major:

  1. Authors should be aware that they do not prepare the basic manual for students, or column of the newspaper, but a scientific paper, so they should (1) use a proper scientific vocabulary and language, (2) avoid excessive simplification. Sentences such as “Adolescents adhere to the JFGST were healthy, and it is useful for adolescents, too.” Are not acceptable.

Response: We have revised the Abstract thoroughly, as well as the whole manuscript, for language and grammar. For example,

“achieving dietary and nutritional balance is an essential goal of JFGST” Lines 71–72

“who had taken breakfast before blood extraction” Lines 99–100

The revised conclusion of the Abstract was the following.

“Our findings suggest that Japanese adolescents adherent to the JFGST had low cardiovascular risks and should not fall lower limits for five food groups.” Lines 24–26

Abstract:

  1. Instead of what was done, Authors should present the aim of the study (e.g. “The aim of the study was…”)

Response: We have corrected the aim of this study in the Abstract, as follows:

“We aimed to investigate whether adherence to the JFGST in 8th grade junior high school students (n = 3162) was associated with cardiometabolic risks and how different scorings of the JFGST influenced the associations.” Lines 14–17

  1. Scoring systems should be briefly described.

Response: We have revised the Abstract, and the scoring systems were described, as follows:

”The Japanese Food Guide Spinning Top (JFGST) indicates optimal intake of five food dishes (grain, fish and meat, vegetables, milk, and fruits) and sugar and confectionaries.” Lines 13–15

“Three types of scoring adherent to the JFGST were analyzed (10 points were given for each item with optimal intake; range: 0–60): the original scoring (ORG scoring), first modified scoring which had no upper limits for vegetables and fruits (MOD1 scoring), and MOD2 scoring without upper limits for five dishes (MOD2 scoring).” Lines 18–21

Introduction:

  1. Instead of what was done, Authors should present the aim of the study (e.g. “The aim of the study was…”)

Response: We have corrected the aim of this study in the Introduction.

“The aim of the study was to investigate the association between diet quality score of adherence to the JFGST, representing a feature of Japanese diet, and metabolic risk factors in Japanese adolescents.” Lines 89–91

Materials and Methods:

  1. The representativeness of the studied group should be reflected.

Response: A total of 6805 students who attended one of the 17 junior high schools of Shunan City between 2006 and 2010 participated in the study. We have added the following sentence to the Materials and Methods section.

“The participants were 8th-grade junior high school students from 17 junior high schools in Shunan City, Japan. From 6805 students attending any of the schools between 2006 and 2010, 6226 students participated in this study with their guardian’s consent.” Lines 96–98

  1. It seems that Authors did not verify the normality of distribution and they treated all the variables as normally distributed, but based on the presented mean and SD values, it may be supposed that data for some variables were characterised by the distribution different than normal.

Authors should (1) verify the normality of distribution, (2) for normally distributed data present mean and SD values, but for the other distributions – present median, min and max values, (3) apply adequate statistical tests, that are based on the distribution.

Response: We have examined the normality using qq plots because the Shapiro-Wilk, and Kolmogorov-Smirnov tests was inappropriate due to the large sample size, or ties. After checking, JFGST score points and triglyceride levels are now expressed as median (min, max). The distribution of the MS score is shown in Table 1, which resembles a Poisson distribution. Other variables were considered to have normal distributions and there was no difference in distributions between before and after log-transformation. In the linear model, triglyceride levels were log-transformed. We have added the following sentences.

“After checking for normality using quantile-quantile plots, the JFGST score points and triglyceride levels are expressed as median (minimum, maximum).” Lines 170–171

“When the triglyceride level was used as a dependent variable, it was natural log transformed.” Lines 179–180

Results:

  1. For normally distributed data Authors should present mean and SD values, but for the other distributions – present median, min and max values.

Authors should apply adequate statistical tests, that are based on the distribution.

Response: According to the corrected methods, relevant results have been revised. However, the overall results and conclusion did not change.

We have added the median (min, max) of triglyceride levels in Table 1.

Triglyceride, mg/dL

52 (15, 249)

In footnotes, “Data are expressed as … median (minimum, maximum).”

We have revised the score points in Lines 199–206 and Table 2.

“Therefore, the median points for fish and meat dishes and milk using the scoring with the upper limits (6.7 and 5 points, respectively) increased when using the scoring without the upper limits (10 and 10 points, respectively; Table 2). For less than 10% of the students, the points for grain dishes (3.4%), vegetable dishes (9.7%), and fruits (4.3%) were above the upper limits. The points for these dishes were similar regardless of whether the upper limits were applied or not. The median JFGST scores were 32 (minimum: 6.7, maximum: 58.3) using the ORG scoring, 33 (3.7, 58.8) using the MOD1 scoring with no upper limits for vegetable dishes and fruits, and 40.9 (15, 60) using the MOD2 scoring with no upper limits for all dishes except for sugar and confectionaries.” Lines 196-204

The results of regression analysis for triglyceride levels in Table 5 has been revised.

Discussion:

  1. Authors should formulate implications of the results of their study and studies by other authors, but they should be relevant not only for Japanese readers, but rather for international auditory

Response: We have added the merit of use of the FOOD-BASED JFGST for Japanese in comparison with existing dietary indices. Food-based dietary indices may be easily used for health promotion.

“Similar to previous dietary indices associated with cardiovascular diseases rather than cancers [1, 2], the JFGST was associated with cardiometabolic risk factors. The JFGST was developed in consideration of the concept of Japanese culture. Japanese terms for dishes are derived from ichiju-sansai, a common formula of Japanese cuisine, Washoku. Grain dishes, shushoku, are the main staples; fish and meat dishes, shusai, are the main dishes; and vegetable dishes, fukusai, are the side dishes [40]. The concept of Japanese dishes is disseminated through homemaking classes and nutrition education at school. The JFGST is a food-based index unlike most of other existing dietary indices [41]. The Healthy Eating Index, the Diet Quality Index, and the Mediterranean Diet Quality Score use foods and nutrients together as scoring items [3, 4, 5]. The Mediterranean Diet Quality Index for children and adolescents uses food intake and dietary habits, but this scoring may be influenced by culture [6]. When using the JFGST as a health promotion and education tool, adolescents can easily understand their suitable dish servings. For effective nutrition education, the dietary index should be suited to the background of the target population.” Lines 317–330

  1. Authors should formulate the future areas which should be studied (not only in Japan, but in various countries)

Response: We have added the future areas of research and implications of the findings in the Discussion section.

“In contrast to the scoring items, the JFGST scoring uses a combination of nutrient and food intake; the servings of grain dishes, fish and meat dishes, and milk were calculated from nutrient amount, but the servings of vegetable dishes and fruits were from the food amount. It is unknown whether servings that adolescents have in mind coincides with those of the JFGST scoring. Dietary indices have three objectives: to measure absolute diet quality, to evaluate adherence to dietary guidelines, and to guide health promotion [41]. According to the third objective, Japanese youths should be educated on favorable dietary habits to ensure longevity and low mortality. Practical use of existing dietary indices for health promotion should be ad-dressed [1, 2], and JFGST should be adopted for nutrition education interventions.” Lines 332–341

Conclusions:

  1. This section is not a summary – Authors should not reproduce the previous sections in this section. Authors should briefly (2-3 sentences) formulate the most important general conclusions based on the conducted study.

Response: We have revised the Conclusion concisely.

“We showed for the first time that Japanese adolescents adherent to the JFGST had low cardiovascular risk. However, only modified JFGST scoring system with no upper limits for five food groups (grain dishes, fish and meat dishes, vegetable dishes, milk, and fruits) was beneficial in alleviating cardiovascular risks; it indicates that adolescent should not fall lower limits of five food group intake. The JFGST can be used for health education of adolescents.” Lines 357–362

Author Contributions:

  1. Authors should be consistent with Authorship (is it FA, or AF?)

Response: We have corrected it. “AF” is the correct format. Line 367

Reviewer 2 Report

This is a comprehensive review on a population which is rather young to adress cardiometabolic risk factors. Those usually emerge after the 3rd decade. But the study is well done and certainly of some interest for the Japanese population 

Author Response

Thank you for your kind review.

Round 2

Reviewer 1 Report

The manuscript entitled „Adherence to the Japanese Food Guide: The association between three scoring systems and cardiometabolic risks in Japanese adolescents” presents interesting issue.

Materials and Methods:

The representativeness of the studied group should be reflected. Representativeness is comparison of the general characteristics of the studied group with the general characteristics of the population (e.g. gender proportions and other basic characteristics in the studied age group).